# Improving Medical Student Anatomy Knowledge and Confidence for the Breast Surgical Oncology Rotation

**DOI:** 10.3390/healthcare11050709

**Published:** 2023-02-27

**Authors:** Chloe Wilder, Lyndsey J. Kilgore, Abbey Fritzel, Kelsey E. Larson

**Affiliations:** 1School of Medicine, University of Kansas Medical Center, Kansas City, KS 66160, USA; 2Breast Surgery Division, Department of Surgery, University of Kansas Medical Center, Kansas City, KS 66160, USA

**Keywords:** anatomy, medical education, surgery education, breast surgery, dissection

## Abstract

Background: The anatomy curriculum has undergone considerable reductions in class time, resulting in decreased student anatomical knowledge retention and confidence during their surgical rotations. To counter this deficit in anatomy knowledge, a clinical anatomy mentorship program (CAMP) was developed by fourth-year medical student leaders and staff mentors in a near-peer teaching fashion prior to the surgical clerkship. This study analyzed the impact this program had on third-year medical students (MS3s) self-assessed anatomical knowledge and confidence in the operating room on the Breast Surgical Oncology rotation after this near-peer program. Methods: A single-center prospective survey study was performed at an academic medical center. Pre- and post-program surveys were administered to all students who participated in the CAMP and rotated on the breast surgical oncology (BSO) service during the surgery clerkship rotation. A control group of individuals who did not rotate on the CAMP was established, and this group was administered a retrospective survey. A 5-point Likert scale was used to assess surgical anatomy knowledge, confidence in the operating room, and comfort in assisting in the operating room. Control group versus post-CAMP intervention group and pre- versus post-CAMP intervention groups survey results were compared using the Student’s t-test with a *p*-value of <0.05 statistically significant. Results: All CAMP students rated their surgical anatomy knowledge (*p* < 0.01), confidence in the operating room (*p* < 0.01), and comfort in assisting in the operating room (*p* < 0.01) as greater than those who did not participate in the program. Additionally, the program improved the ability of third-year medical students to prepare for operating room cases going into their third-year breast surgical oncology clerkship (*p* < 0.03). Conclusions: This near-peer surgical education model appears to be an effective way to prepare third-year medical students for the breast surgical oncology rotation during the surgery clerkship by improving anatomic knowledge and student confidence. The program can serve as a template for medical students, surgical clerkship directors, and other faculty interested in efficiently expanding surgical anatomy at their institution.

## 1. Introduction

Historically, a deep understanding of anatomy to diagnose and treat patients was considered the cornerstone of medicine. However, in recent decades, a drastic shift has occurred so that anatomy is now only a small part of modern medical education [1,2]. Reports from the early 1900s indicate that American medical students spent over 800 hours mastering anatomy [3]. In the 1950s, the number of hours had dropped to below 350 with recent data suggesting students now spend less than 150 hours in the anatomy lab [2,4]. Furthermore, anatomy is often presented in self-directed and team-based learning models, which lack cadaveric dissection [5,6,7]. Shifting the emphasis away from cadaver lab experience has led to decreased anatomical knowledge amongst medical school graduates [8,9]. A survey of general surgery residency program directors demonstrated that over half of the program directors believed incoming residents were less prepared than residents ten years prior [9]. Medical students agree that anatomy teaching is not meeting their needs and feel inadequately prepared to use their knowledge of anatomy in practice [10].

Medical educators agree that anatomy must remain a core subject, as physicians must be experts in distinguishing normal versus pathologic variants, a skill that begins with a foundation in anatomic knowledge [11]. Without proper anatomical knowledge, surgeons may be insufficient at investigating and intervening on behalf of their patients [12]. In addition, anatomic errors are linked to increases in financial and litigious claims. The increase in litigious claims is related to some errors involving surrounding structures, pointing towards an inadequate understanding of anatomic relationships when performing procedures [13]. 3D spatial reasoning is key when evaluating patient anatomical pathology, a process that begins with cadaveric dissection [14].

Another issue that presents itself is knowledge retention. Preclinical students are known to score better on anatomical tests than surgical specialists [15]; however, there is a 50% drop in knowledge retention when first-year medical students begin their clinical 3rd-year rotations [16]. Even junior doctors, who have recently graduated from medical school, are underperforming when tested on anatomy [17]. These alarming results outline the issue of clustering anatomy education in the preclinical years and expecting students to retain or, at some level, regain anatomy sufficiency during clinical training. Ultimately, if anatomical knowledge is regained but not retained during rotations [16], this further stresses the critical need for repetition to ensure retention [18].

When asked about their confidence in anatomy or their students’ anatomy learning, students and faculty both express a desire for more time spent with cadavers and a longer timeline for learning [10,19]. They also agree that the best way to learn during these cadaver prosections is in small student groups with qualified demonstrators [20]. Other studies that implemented courses during or after clinical rotations reported increases in test scores when identifying anatomical structures and student confidence [21,22]. More interesting is the idea that even a short refresher course is enough to increase scores greatly [23,24]. Mid-course scores are not significantly improved from post-course scores, indicating that extra, post-preclinical anatomy courses do not need time from other curriculum areas to be effective. Vertical integration of continuing anatomy education throughout all years of the medical school curriculum is desired and proven to increase retention [10,24,25].

Combining basic science anatomical content within clinical disciplines can enhance the retention of both core subjects [26]. While this relationship is detailed, some faculty view preclinical basic science separately from clinically oriented anatomy and, therefore, teach it outside of the realm of functional understanding [27]. Clinically relevant procedure rehearsals through small-group prosection initiate the process of surgical learning by reducing cognitive load without the added stress of patient outcomes [28]. The transfer of anatomical knowledge past superficial memorization is increased by learning from kinetic examples in abstract problem-solving [29]. This is exactly what small-group prosection in a clinical orientation led by qualified demonstrators aims to achieve.

Near-peer teaching has been explored as a way for both tutor and tutee to become more confident when learning complex clinical anatomy [30]. Therefore, students who receive proper training on how to teach a clinical topic become qualified demonstrators, feel more confident in themselves [31,32], and are respected as a reputable source of information from their tutees [33]. Having an expert in the room to ask questions that the near-peer teachers may not know is recognized as necessary [33,34]. However, peer teaching frees up many faculty to facilitate in-depth questions for larger groups of students while providing students an environment in which they are more comfortable engaging and asking questions of their peers.

Since medical students spend less time in the anatomy lab, the Clinical Anatomy Mentorship Program (CAMP) was created to address these issues through a student-led initiative supervised by general surgery faculty. The CAMP details common operations to improve surgical anatomy confidence and spatial awareness through a clinical lens. The CAMP breast surgical oncology (BSO) prosection allows visuospatial familiarization with tangible anatomy while exposing third-year medical student (MS3) learners to the clinical functionality of the breast procedure in a low-stress, peer-teaching environment. This nested teaching of concrete examples with clinical relevance to a wider understanding of global anatomy leads to increased confidence in the learners and potentially longer retention of basic science knowledge rooted in clinical application.

In this publication, we focused specifically on the BSO CAMP curriculum. We aimed to determine the impact of the curriculum on the students’ self-assessed anatomical knowledge and confidence in the operating room (OR) by comparing students who did and did not experience the CAMP education. We hypothesized that the BSO CAMP would improve the students’ knowledge and confidence for their subsequent MS3 surgical rotation.

## 2. Materials and Methods

This single-institution study was approved by the Institutional Review Board (IRB). A list of MS3s who had completed the BSO rotation from June 2019 to May 2020 (prior to the implementation of the BSO CAMP education) was obtained from the clerkship director. This MS3 cohort served as the control group and was issued a retrospective, anonymous, and voluntary online survey designed to assess their knowledge and confidence during their MS3 surgery rotation. A second list of MS3s rotating on the BSO rotation during the surgery clerkship was obtained from the clerkship director prior to each eight-week clerkship from June 2020 to April 2021 in a prospective manner. This group served as the intervention group. All students assigned to the BSO rotation were invited to attend the BSO CAMP session prior to beginning their surgery clerkship and were issued anonymous pre- and post-CAMP surveys.

### 2.1. Establishing Baseline Data (Control Group)

The control group survey explored the following themes: surgical anatomy knowledge, confidence in answering questions in the OR, comfort in assisting in the OR, and ability to prepare for surgical cases (Figure 1). Students were asked to give a rating in each of these areas based on a 5-point Likert rating scale (very poor, poor, fair, good, excellent). Students were asked to respond to these self-assessment questions as reflected at the end of their BSO rotation. The survey was issued in a retrospective fashion at a single point in time (June 2020); as a result, some students were closer versus further removed from their rotation.

### 2.2. Curriculum Details

The primary teachers for the BSO CAMP were fourth-year medical students (MS4s) who had previously completed the BSO rotation. These individuals served as near-peer teachers for their MS3 colleagues prior to the start of their surgical rotation. During the CAMP sessions, the MS3 students worked directly with the MS4 teachers to review the curriculum, which included two clinical cases in conjunction with a prosection anatomy review of two common surgical cases on one Thiel-embalmed cadaver. In total, the students spent 1 hour reviewing the cases and procedures detailed below.

The reference material was written by the breast surgical oncology faculty in a stepwise fashion to walk students through the patient evaluation and subsequent operation in an organized manner, mirroring what they would encounter on their surgical rotation. The two key surgical procedures were mastectomy and axillary lymph node dissection. The students were asked to identify critical anatomic landmarks on the prosection, key steps in the operation, and relevant clinical or pathophysiologic correlations at each step as the central learning objectives.

### 2.3. BSO CAMP Student Survey (Intervention Group)

The MS3 students enrolled in the BSO rotation attended a breast surgery specific CAMP teaching session for an hour during the didactic week prior to starting their surgery clerkship. Before attending the CAMP session, the MS3s were issued a pre-CAMP survey, which mirrored the control survey (Figure 2). The students then attended the BSO CAMP teaching session as described above. After completing their BSO surgery rotation, the students were issued a post-CAMP survey (Figure 3), which contained supplemental questions assessing the influence of the BSO CAMP on their surgery rotation experience.

### 2.4. Statistical Analysis

The survey data were securely stored in an institutional secure online REDCap database. During analysis, de-identified survey responses were converted to numerical data points based on a 5-point Likert scale (1-very poor, 2-poor, 3-fair, 4-good, 5-excellent) and agreement (1-strongly disagree, 2-disagree, 3-neutral, 4-agree, 5-strongly agree). The responses for each survey question were averaged for the purposes of analysis and reported with the standard deviation. Normality testing was not performed, and we assumed a normal distribution in our analysis. Control group versus post-CAMP intervention group and pre- versus post-CAMP intervention groups survey results were compared using the Student’s t-test with a *p*-value of <0.05 statistically significant.

The validity argument for the items used in the survey include our teams’ inferences and assumptions about the survey items. In particular, the validity arguments involved assumptions about how to best measure for the variables of interest in a study [35]. It was critical to our research that we measured the students’ subjective experiences. For example, we did not intend to objectively measure the students’ anatomy knowledge, rather we wanted to focus on the learners’ self-assessment of their knowledge, confidence, comfort, and ability to prepare for operative cases. The surveys assess one item per domain to efficiently measure the variables of interest and are more appropriate than other methods of data collection.

## 3. Results

All eligible students who were invited to participate completed the control (n = 9), pre-CAMP (n = 11), and post-CAMP surveys (n = 11). Figure 1 compares the control group (students who did not experience the CAMP) to the intervention group (separated into pre- and post-CAMP). Table 1 details the Likert scale responses between the three groups and their associated *p*-values. The top *p*-value is measured between the control group and the post-CAMP intervention group, the bottom *p*-value is measured between the pre- and post-CAMP intervention groups.

The first survey question (Q1 in Figure 2, Figure 3 and Figure 4) sought to examine the MS3s’ surgical anatomy knowledge. Prior to completing the BSO CAMP, the majority of the intervention group ranked their anatomy knowledge as poor (55%) or very poor (18%) (mean (SD) 2.18 ± 0.87). After participating in the BSO CAMP, 100% of the intervention group ranked their knowledge as good (45%) or excellent (55%) (*p* = 0.0001, mean (SD) 4.5 ± 0.52). The MS3s from the control group ranked their anatomy knowledge lower than those who had completed the BSO CAMP (*p* = 0.0001, mean (SD) 3.67 ± 0.5 vs. 4.36 ± 0.67). Following the CAMP, 73% strongly agreed, and 27% agreed that the BSO CAMP improved their surgical anatomy knowledge (mean (SD) 4.73 ± 0.47).

The second question (Q2 in Figure 2, Figure 3 and Figure 4) explored confidence in answering anatomy-based questions in the operating room. Students in the intervention group, prior to the CAMP, ranked their confidence in answering questions in the OR as fair (18%), poor (72%), or very poor (10%) (mean (SD) 2.09 ± 0.54). After the CAMP, the confidence of the MS3s increased greatly compared to the pre-CAMP levels (*p* = 0.0046, mean (SD) 4.36 ± 0.67 vs. 2.09 ± 0.54). The intervention group’s confidence was greater than the control group who learned from the clinical rotation alone (*p* = 0.0001, mean (SD) 4.37 ± 0.67 vs. 3.11 ± 1.05). In the intervention group, all respondents strongly agreed (82%) or agreed (18%) that the BSO CAMP improved their confidence in answering questions in the operating room (mean (SD) 4.81 ± 0.40).

The third question (Q3, Figure 2, Figure 3 and Figure 4) sought to analyze the MS3s’ comfort in assisting surgical cases, which was considered related to anatomic and surgical case knowledge. The majority of the control group ranked their comfort assisting as good (56%) or worse (mean (SD) 3.44 ± 0.73). In contrast, all the MS3s in the intervention group felt their comfort assisting in surgical cases was good (73%) or excellent (27%) post-CAMP, a significant improvement from their pre-CAMP ranking (*p* = 0.0002, mean (SD) 4.27 ± 0.47 vs. 2.73 ± 1.01). The majority of students agreed (64%) or strongly agreed (18%) that the BSO CAMP improved their comfort in assisting in surgical cases (mean (SD) 4.0 ± 0.63).

The fourth theme (Q4, Figure 2, Figure 3 and Figure 4) explored preparedness, defined as the ability to gather relevant information about the patient and surgery prior to the start of the procedure. The majority (53%) of the control group felt their preparedness was good or excellent, similar to the post-CAMP group (*p* = 0.06, mean (SD) 4.22 ± 0.67 vs. 4.73 ± 0.47). The CAMP did significantly improve the students’ preparedness (*p* = 0.003, mean (SD) pre-CAMP 3.18 ± 1.08 vs. post-CAMP 4.73 ± 0.47).

## 4. Discussion

As part of the MS3 clerkship education, the BSO CAMP improves medical student confidence, comfort, and ability to prepare for the surgical rotation. The BSO CAMP curriculum focuses on providing students the tools to be successful in their surgical rotation, including how to learn anatomy and not the primary retention of specific anatomy. Notably, most students attributed their improvement to completing the CAMP curriculum. The improvement in rankings and attribution of improvement to the curriculum demonstrate that the BSO CAMP was successful in its goal as an anatomy educational tool. The results of our study may be applicable to other medical schools looking to expand their clinical anatomy teaching and surgical curriculum.

Our control cohort survey confirmed that the students felt that their anatomy knowledge, confidence, and comfort were less than good, reflecting prior students’ self-assessments from other institutions. In particular, our control data is consistent with an earlier study performed by Fitzgerald et al. [10], where graduating medical students surveyed felt that they received insufficient anatomy instruction during their training. In this prior publication, nearly half of the surveyed students felt they had not received adequate anatomy teaching when departing medical school; our control group data support this same concern, reflecting a common theme across institutions. Our data reaffirms the critical need for ongoing anatomy education such as the CAMP in the medical school curriculum.

The most significant gap in educational need appears to be between preclinical and clinical years. Prior studies demonstrated that medical students entering their clinical years are ill-prepared to transfer anatomical knowledge to practice and suggested a need for anatomy courses coinciding with clinical education [16,36]. However, there is a paucity of data on educational endeavors to address this need. A limited number of schools have implemented clinical anatomy electives during the MS4 year with significant improvement in anatomic knowledge documented following the elective [37,38,39]. Unfortunately, courses such as these may be designed specifically for MS4s entering surgery rather than being available to all medical students [39]. The BSO CAMP program evaluated in this publication is open to all MS3 students regardless of potential specialty, reflecting a different timing and broader audience for our curriculum versus those previously reported.

The BSO curriculum provides an opportunity for vertical integration within the medical school curriculum. Medical education strives to integrate clinical topics into preclinical years, but it does not always integrate basic science principles into the clinical years, creating a unidirectional system [24,40]. Opportunities such as the BSO CAMP allow the integration of preclinical anatomy principles to be re-introduced at appropriate and relevant times in the clinical curriculum. We demonstrated that utilizing the surgery clerkship is a potentially beneficial time to review anatomy knowledge during the clinical years. The proximity of the BSO CAMP anatomy review to the breast surgery rotation allows students to contextualize and further consolidate the information being taught in the classroom. The improvements in the MS3 survey results suggest that this approach was a positive adjunct to the clerkship and a beneficial use of the students’ time.

In developing the BSO CAMP curriculum, we combined results from our control cohort surveys with recurring themes from similar studies to structure the educational content so that it met the needs of senior medical students [9,30]. Common and reoccurring themes include the need to teach clinically oriented anatomy, emphasizing the need for anatomy courses taught by qualified demonstrators (those skilled at anatomic knowledge in a clinical context), and the need for refresher courses to highlight forgotten knowledge. From this data, we created and implemented a BSO CAMP curriculum that met three learner-directed goals: (1) teach clinically oriented anatomy from a surgical perspective, (2) teach in close proximity to the surgical rotation for better consolidation of concepts, and (3) utilize MS4 mentors as teachers for near-peer anatomy review.

One challenge in developing the CAMP was to identify the hands-on learning approach that best fit our students’ needs. Prior authors have widely differing proposals for how to teach anatomy in the modern era, including abandoning gross anatomy labs in favor of simulation, virtual reality, and clinical skills-based learning [11,24]. Prior publications assessing prosections note the strength of this approach as more time-efficient and cost-effective than dissection with the benefit of fewer cadavers required [41]. However, prosections as a means of anatomy education are generally utilized during MS1 and MS2 years only. To our knowledge, applying this concept to the MS3s’ surgery clerkship has not robustly been described in the literature.

While the positive MS3 survey results presented here are encouraging, there are several limitations to consider. First are those that relate to the CAMP curriculum itself. There are potential issues regarding the use of cadaveric dissection to teach anatomy, including expense, resource limitations, and time [11,42]. In the CAMP, each of these concerns was addressed individually. Theil-embalmed cadavers are utilized as they allow prosections to be used for repeated teaching sessions over a period of several months; this saves total cost over the year despite each individual cadaver costing more [43]. Another finite resource is the surgical faculty’s time for education. By using peer mentors (MS4s), surgical faculty were present to confirm the MS4s’ knowledge and prosection anatomy validity, but the MS4s led ongoing, longer, and more frequent MS3 teaching sessions throughout the year. In addition, time is also a valuable resource for medical students and the medical school curriculum. Therefore, the focus of the CAMP was specialty-specific rather than generalized. Providing a short, high-yield session prior to the clerkship allowed appropriate MS3 education while still protecting the time of the students participating.

Second, with a small sample size surveyed in our study, it is unclear if the data is widely applicable to MS3s in other hospitals or across classes at our institution. While this is a limitation in our study’s initial survey configuration (using a Likert scale), it can be remedied in the future with larger cohorts. We will continue collecting data prospectively to determine if the positive educational experience is reproducible for ongoing years. Third, our study design did not have a control group in real-time but rather a retrospectively surveyed control group. This design was intentional, as we did not wish to exclude any student from the CAMP as an educational opportunity. This does not negate our results, but it is noted as a potential confounding factor.

Finally, we recognize that objective measurements of anatomy knowledge and retention are widely used as a concrete baseline of student learning. However, our study focused on the subjective experience of the medical students and their self-assessment of knowledge, confidence, and preparedness. Although our study did not investigate anatomy education in a concrete sense, student metacognition of the practices surrounding surgery can still be indicative of overall success. In the future, correlating the students’ experience to scores can be beneficial to concretely make the association between positive self-assessments and higher scores.

Despite these limitations, our results suggest that implementing the BSO CAMP curriculum improved the MS3 surgery clerkship. Based on our findings, MS3 education may benefit from a vertical integration model such as the BSO CAMP to improve surgical anatomy knowledge. Focusing content on high-yield anatomy and scheduling the session to coincide with the surgery clerkships are beneficial and efficient based on our results. Going forward, medical schools could consider the CAMP as an educational model useful for integrating anatomy and clinical concepts for MS3s.

## 5. Conclusions

The CAMP BSO for MS3 students effectively prepares students and improves confidence prior to the breast surgical oncology clerkship rotation. Using the CAMP via near-peer mentor teaching, the students improved their knowledge, confidence, comfort, and ability. This model can serve as a template for medical students, surgical clerkship directors, and other faculty interested in expanding surgical anatomy education at their institution.

## Figures and Tables

**Figure 1 healthcare-11-00709-f001:**
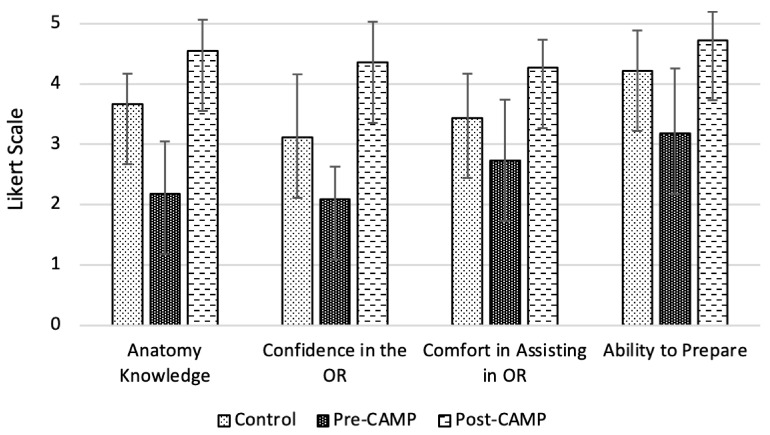
Summary of MS3 responses to the survey items, including the control group and intervention group pre- and post-CAMP. The data is averaged.

**Figure 2 healthcare-11-00709-f002:**
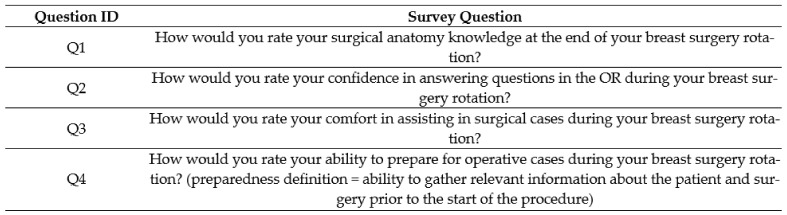
Survey for the control group administered to those who had completed the BSO clerkship but never participated in the CAMP. Responses were given on a 5-point Likert scale ranging from very poor to excellent.

**Figure 3 healthcare-11-00709-f003:**
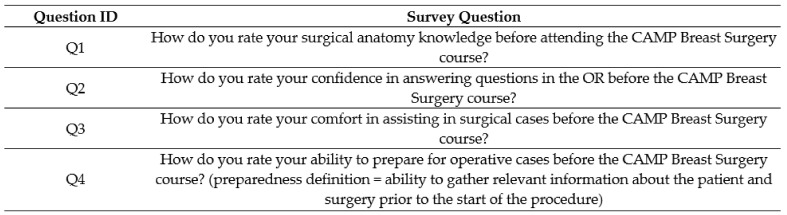
Pre-CAMP survey to be taken by the intervention group before the CAMP teaching session. Responses were given on a 5-point Likert scale ranging from very poor to excellent.

**Figure 4 healthcare-11-00709-f004:**
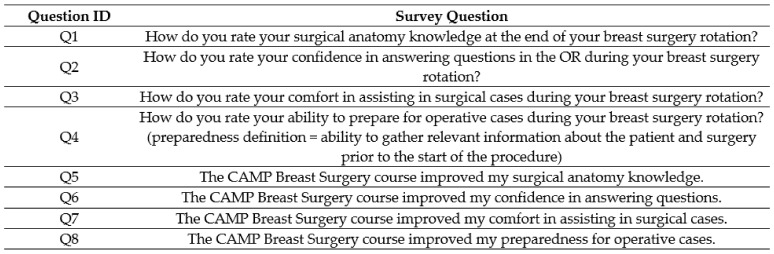
Post-CAMP survey to be taken by the intervention group after the CAMP teaching session. All responses were given on a Likert scale ranging from very poor to excellent (Q1–Q4) or strongly disagree to strongly agree (Q5–Q8).

**Table 1 healthcare-11-00709-t001:** Likert Scale Average Response (SD) for All Cohorts.

Question ID	CONTROL (n = 9)	PRE-CAMP (n = 11)	POST-CAMP (n = 11)	*p*-Value *
Q1. Anatomy Knowledge	3.67 (0.5)	2.18 (0.87)	4.55 (0.52)	0.0001 * 0.0001 *
Q2. Confidence	3.11 (1.05)	2.09 (0.54)	4.36 (0.67)	0.0046 * 0.0001 *
Q3. Comfort	3.44 (0.73)	2.73 (1.01)	4.27 (0.47)	0.0065 * 0.0002 *
Q4. Preparation	4.22 (0.67)	3.18 (1.08)	4.73 (0.47)	0.06 0.003 *
Q5. CAMP impact on anatomy			4.73 (0.47)	
Q6. CAMP impact confidence			4.81 (0.40)	
Q7. CAMP impact comfort			4.0 (0.63)	
Q8. CAMP impact preparation			4.36 (1.03)	

* *p*-value <0.05 statistically significant. 1st line of *p*-value represents control versus post-CAMP. 2nd line of *p*-value represents pre- versus post-CAMP.

## Data Availability

Data is unavailable due to privacy restrictions.

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
