# Peer review of "Improving Medical Student Anatomy Knowledge and Confidence for the Breast Surgical Oncology Rotation"

_healthcare, 2023, doi:10.3390/healthcare11050709_

Round 1
Reviewer 1 Report
It is an interesting manuscript that exposes the importance of anatomy in the programming of surgical teaching. This is important because it fulfills two objectives of teach clinically oriented anatomy from a surgical perspective and teach in close proximity to the surgical rotation for a better consideration of concepts. The teaching of anatomy must be continuous throughout the curriculum.
Author Response
Thank you for your comments regarding our paper, Improving Medical Student Anatomy Knowledge and Confidence for the Breast Surgical Oncology Rotation. We appreciate your positive feedback.
Reviewer 2 Report
Thank you for the opportunity to review this paper on the effect of a Clinical Anatomy Mentorship Program (CAMP) for medical students in a breast cancer surgery rotation (BSO). The paper describes a rather small study with 11 participants in the intervention group. The students in the intervention went through CAMP. The control group was BSO students from the previous year (n=9). The effect of the intervention was assessed pre and post-BSO in the intervention group and post-BSO in the control group.
- The study is limited in size, and there are methodological issues regarding the instrument used. However, the study has a feasibility approach, is well-written and warrants publication.
- CAMP needs to be more thoroughly described. What did the CAMP consist of? How much time did each student spend with their CAMP mentors? Was it just one demonstration on a prepared cadaver or several demonstrations? What were the learning objectives? Costs? Any measurement of what was actually learned?
- The statistics would need to be done using non-parametric methods as n is small, and the Likert scale is not a parametric scale.
- What are the arguments for the validity of the used items, and why do you have one item per domain?
- The items used are on a low level on Kirkpatrick's scale as they measure the students' feelings towards what they have learnt. In modern educational literature, this is an unreliable way to assess education. This would need to be addressed in limitations.
- The information about the institutional review board would need to be supplemented with information on the date of the decision and the chair of the Board or a document number.
Author Response
REVIEWER 2 COMMENT: CAMP needs to be more thoroughly described. What did the CAMP consist of? How much time did each student spend with their CAMP mentors? Was it just one demonstration on a prepared cadaver or several demonstrations? What were the learning objectives? Costs? Any measurement of what was actually learned?
RESPONSE TO REVIEWER 2: Within the materials and methods, Curriculum Details, section there has been added a descriptor of how much time the students spent doing the two clinical cases and the two key surgical procedures (mastectomy and axillary lymph node dissection). We have added clarity in lines 142-143 of procedures to cadaver ratio. The two demonstrations are detailed in line 148. Lines 149-151 have been specified as the key learning objectives.
Discussion of costs of the CAMP curriculum including negotiated costs, in-kind costs, grant and donated funding, and other cost considerations are outside of the scope of this article and confidential financial information for the institution. In the first paragraph of the discussion section we indicate that quantitative measurement of learning was not conducted as this was not the intended goal of CAMP. This is also indicated in the methods section on statistical analysis.
REVIEWER 2 COMMENT: The statistics would need to be done using non-parametric methods as n is small, and the Likert scale is not a parametric scale.
RESPONSE TO REVIEWER 2: Thank you for the feedback. We are unable to go back and adjust the survey, and do not feel it is appropriate to adjust the Likert scale (original survey model) after the survey results were collected. We do agree with your assessment that this is a limitation in selecting this survey option with small numbers. We have added a comment about this to the discussion section to acknowledge your feedback.
REVIEWER 2 COMMENTS: What are the arguments for the validity of the used items and why do you have one item per domain?
RESPONSE TO REVIEWER 2: We have added language to the methods section that summarizes the validity arguments of the items we used in this study.
REVIEWER 2 COMMENTS: The items used are on a low level on Kirkpatrick’s scale as they measure the students’ feelings towards what they have learnt. In modern educational
literature, this is an unreliable way to assess education. This would need to be addressed in
limitations.
RESPONSE TO REVIEWER 2: We have addressed this in both the first paragraph of the discussion and in the methods section. We have clarified the limitations of focusing on the subjective feelings of the students in the 9th paragraph of the discussion section.
REVIEWER 2 COMMENTS: The information about the institutional review board would need to be supplemented with information on the date of the decision and the chair of the Board or a document number.
RESPONSE TO REVIEWER 2: Our institution does not have an IRB chair/board member who is responsible for each individual review. This was classified as quality improvement and thus was deemed IRB exempt, so a documentation or study number is not indicated based on protocol in our center.
Reviewer 3 Report
I enjoyed reading the manuscript because it underscores one of the most exciting topics in the medical education system. The authors describe the shortage in anatomy knowledge which is relative to the reduction of teaching hours which applies to all disciples due to the expansion of the information of each field which is not compensated with the required credit hours in the medical schools' curricula.
They proposed a solution to improve anatomy knowledge to gain competent graduates through a model of BSO-CAMP model.
Despite the low sample size but I think this pilot study can be a raw model for larger studies.
Author Response
Thank you for your comments regarding our paper, Improving Medical Student
Anatomy Knowledge and Confidence for the Breast Surgical Oncology Rotation. We appreciate your positive feedback and insight into the reduction of teaching hours in all disciplines.
Reviewer 4 Report
The authors provide an interesting study investigating the impact of a clinical anatomy mentorship program or CAMP on third year medical students’ self-rated anatomical knowledge and confidence in the OR during their rotation following completion of this program. They found that students who took part in CAMP rated their surgical anatomy knowledge, confidence in the OR and comfort in assisting in the OR significantly higher than those who did not take part in the program.
The background into the study, objectives, discussion are comprehensive and clearly presented. The discussion articulated the importance of adopting programs such as CAMP for medical students, surgical clerkship directors etc in the field of anatomy. I also appreciate the discussion on the limitations of the study, in which the authors acknowledge key setbacks with the study design.
However, there are some concerns regarding the methodology and results sections of the study, as outlined below. If the authors can address these concerns it would make a great and important article for this journal.
· Please include a statement on whether the curriculum for the 2019/2020 control cohort was the exact same as that of the 2020/2021 experimental cohort
· Statistical analysis:
- Please include the normality test performed to show data is normal to conduct t-test.
- An ANOVA is a better test to use rather than doing multiple t-tests. Please also include results for subsequent post hoc tests
- Finally, a comparison of ratings for control v pre-CAMP group should be included as it is previously stated that these surveys mirror each other and more importantly, it would be interesting to see if there are, if any, differences between these two groups
· Figure 1 – please remove the means from the top of the bars as it is already clear from the y-axis. Instead, please add + standard deviation error bars. In the figure legend, please add that the data has been averaged
· In page 5 of the results, I do not understand why the mean rank is being reported. When it was previously stated that t-tests were employed. Mean ranks are used during non-parametric analysis, which t-tests are not – reiterating my previous comment to include whether data follows a normal distribution.
· Lines 187 and 189, p-values should be the other way around. Line 188 mean for post-camp confidence should be 4.36
· Please delete lines 206-207, “…., but compared to the control group, this improvement was similar to the rotation alone, as noted above”. No statistical analysis comparing post-CAMP rating of Q8 to control group conducted, thus cannot conclude this.
· Similarly, please reword lines 219- 220, “Students reported improved scores in all areas compared to both control and pre-CAMP survey data”, as there was no difference in ratings between the control group and post-CAMP on preparedness.
· Please also reword lines 225-226, “Our control cohort survey confirmed that students felt that their anatomy knowledge, confidence, comfort, and preparedness were less than good, …”. Mean rating for preparation was in fact good (4.22). This statement needs amending.
Author Response
REVIEWER 4 COMMENTS: Please include a statement on whether the curriculum for the 2019/2020 control cohort was the exact same as that of the 2020/2021 experimental cohort
RESPONSE TO REVIEWER 4: The 2019/2020 Control cohort did not receive the BSO CAMP extra education. They only completed their Breast Surgical Oncology subspecialty rotation during their surgery rotation without first going through BSO CAMP. We have added clarifying language to the first paragraph of material and methods to reflect this.
REVIEWER 4 COMMENTS: Please include the normality test performed to show data is normal to conduct t-test.
RESPONSE TO REVIEWER 4: While normality testing can be considered it is not standard recommendation or approach at our center or in the literature. In the methods, we have acknowledged that normality testing was not performed and that we have assumed a normal distribution in our analysis based on the reviewer’s comment. “From a formal perspective, preliminary testing for normality is incorrect and should therefore be avoided.” (Rochon et al PMC Med Res Methodology. 12, 81 (2012).
REVIEWER 4 COMMENTS: An ANOVA is a better test to use rather than doing multiple t-tests. Please also include results for subsequent post hoc tests
RESPONSE TO REVIEWER 4: ANOVA is used to compare three groups whereas t-test is used to compared two groups. Our study has two groups (those who participated in CAMP and those who did not) so it would not be appropriate to use ANOVA analysis. Since those in the pre- CAMP and post-CAMP are the same individuals surveyed at different times, this would not be appropriate to apply ANOVA.
REVIEWER 4 COMMENTS: Finally, a comparison of ratings for control v pre-CAMP group should be included as it is previously stated that these surveys mirror each other and more importantly, it would be interesting to see if there are, if any , differences between these two groups.
RESPONSE TO REVIEWER 4: This is shown in figure 1. We have added clarifying language in the first paragraph of the results section.
REVIEWER 4 COMMENTS: Figure 1 – Please remove the means from the top of the bars as it is already clear from the 7-axis. Instead, please add + standard deviation error bars. In the figure legend, please add that the data has been averaged.
RESPONSE TO REVIEWER 4: The graph has been updated and in the note beneath the graph we clarified that the data has been averaged.
REVIEWER 4 COMMENTS: In page 5 of the results, I do not understand why the mean rank is being reported. When it was previously stated that t-tests were employed. Mean ranks are used during non-parametric analysis, which t-tests are not – reiterating my previous comment to include whether data follows a normal distribution.
RESPONSE TO REVIEWER 4: Thank you for noticing this incorrect phrasing. In the charts, the data was labelled correctly (mean (SD)) but in the paragraphs of the results section it was incorrectly labelled as mean rank. This has been updated to appropriately reflect that what is being reported is mean(SD), which is a standard outcome of the t-test.
REVIEWER 4 COMMENTS: Lines 187 and 189, p-values should be the other way around. Line 188 mean for post-camp confidence should be 4.36.
RESPONSE TO REVIEWER 4: This has been reflected in the paper.
REVIEWER 4 COMMENTS: Please delete lines 206-207, “…, but compared to the control group, this improvement was similar to the rotation alone, as noted above”. No statistical analysis comparing post-CAMP rating of Q8 to control group conducted, thus cannot conclude this.
RESPONSE TO REVIEWER 4: This has been reflected in the paper.
REVIEWER 4 COMMENTS: Similarly, please reword lines 219-220, “Students reported improved scores in all areas compared to both control and pre-CAMP survey data”, as there was no difference in ratings between the control group and post-CAMP on preparedness.
RESPONSE TO REVIEWER 4: This has been reflected in the paper.
REVIEWER 4 COMMENTS: Please also reword lines 225-226, “our control cohort survey confirmed that students felt that their anatomy knowledge, confidence, comfort, and preparedness were less than good,…”Mean rating for preparation was in fact good (4.22). This statement needs amending.
RESPONSE TO REVIEWER 4: This has been reflected in the paper.